# Detecting rare diseases in electronic health records using machine learning and knowledge engineering: Case study of acute hepatic porphyria

**Aaron M. Cohen**[1]\*, **Steven Chamberlin**[1], **Thomas Deloughery**[1], **Michelle Nguyen**[1], **Steven Bedrick**[1], **Stephen Meninger**[2], **John J. Ko**[2], **Jigar J. Amin**[2], **Alex J. Wei**[2], **William Hersh**[1]

**1** Department of Medical Informatics & Clinical Epidemiology, School of Medicine, Oregon Health & Science University, Portland, Oregon, United States of America, **2** Alnylam Pharmaceuticals, Cambridge, Massachusetts, United States of America

\* cohenaa@ohsu.edu

**Data Availability Statement:** The source data used for this project is electronic health record (EHR) data, and contains protected health information

## Abstract

### Background

With the growing adoption of the electronic health record (EHR) worldwide over the last decade, new opportunities exist for leveraging EHR data for detection of rare diseases. Rare diseases are often not diagnosed or delayed in diagnosis by clinicians who encounter them infrequently. One such rare disease that may be amenable to EHR-based detection is acute hepatic porphyria (AHP). AHP consists of a family of rare, metabolic diseases characterized by potentially life-threatening acute attacks and chronic debilitating symptoms. The goal of this study was to apply machine learning and knowledge engineering to a large extract of EHR data to determine whether they could be effective in identifying patients not previously tested for AHP who should receive a proper diagnostic workup for AHP.

### Methods and findings

We used an extract of the complete EHR data of 200,000 patients from an academic medical center and enriched it with records from an additional 5,571 patients containing any mention of porphyria in the record. After manually reviewing the records of all 47 unique patients with the ICD-10-CM code E80.21 (Acute intermittent [hepatic] porphyria), we identified 30 patients who were positive cases for our machine learning models, with the rest of the patients used as negative cases. We parsed the record into features, which were scored by frequency of appearance and filtered using univariate feature analysis. We manually choose features not directly tied to provider attributes or suspicion of the patient having AHP. We trained on the full dataset, with the best cross-validation performance coming from support vector machine (SVM) algorithm using a radial basis function (RBF) kernel. The trained model was applied back to the full data set and patients were ranked by margin distance. The top 100 ranked negative cases were manually reviewed for symptom complexes similar

(PHI) for patients under care at Oregon Health & Science University (OHSU). The OHSU Institutional Review Board (IRB) does not allow release of this data to the public, and doing so would violate US HIPAA laws. The OHSU IRB can be contacted at: irb@ohsu.edu. Questions about data requests may be sent to this address. We are including full details of the machine learning model, training methods, and final features. Other investigators experienced in the field should be able to reproduce our methods on their own data to validate the results presented in this manuscript.

**Funding:** AMC, BH, SC, and MN received support for this work from Alnylam Pharmaceuticals, Inc., Cambridge, MA. SM, JJK, JJA and AJW are/were employees of Alnylam Pharmaceuticals, Inc., Cambridge, MA during the time of this research. This work was funded and the associated editorial support was provided by Alnylam Pharmaceuticals, Inc., Cambridge, MA. Grant number 4510005336 https://www.alnylam.com/ Alnylam participated in algorithm design and preparation of the manuscript. They had no role in the evaluation or EHR data collection and analysis, nor did they have any access to the individual patient electronic health record data used in this research.

**Competing interests:** I have read the journal's policy and the authors of this manuscript have the following competing interests: GIVLAARI is a product of Alnylam. GIVLAARI is a prescription medicine used to treat acute hepatic porphyria (AHP) in adults. This work was funded and the associated editorial support was provided by Alnylam Pharmaceuticals, Inc., Cambridge, MA. Grant number 4510005336, https://www.alnylam.com/ Alnylam participated in algorithm design and preparation of the manuscript. They had no role in the evaluation or EHR data collection and analysis, nor did they have any access to the individual patient electronic health record data used in this research. This does not alter our adherence to PLOS ONE policies on sharing data and materials.

to AHP, finding four patients where AHP diagnostic testing was likely indicated and 18 patients where AHP diagnostic testing was possibly indicated. From the top 100 ranked cases of patients with mention of porphyria in their record, we identified four patients for whom AHP diagnostic testing was possibly indicated and had not been previously performed. Based solely on the reported prevalence of AHP, we would have expected only 0.002 cases out of the 200 patients manually reviewed.

## Conclusions

The application of machine learning and knowledge engineering to EHR data may facilitate the diagnosis of rare diseases such as AHP. Further work will recommend clinical investigation to identified patients' clinicians, evaluate more patients, assess additional feature selection and machine learning algorithms, and apply this methodology to other rare diseases. This work provides strong evidence that population-level informatics can be applied to rare diseases, greatly improving our ability to identify undiagnosed patients, and in the future improve the care of these patients and our ability study these diseases. The next step is to learn how best to apply these EHR-based machine learning approaches to benefit individual patients with a clinical study that provides diagnostic testing and clinical follow up for those identified as possibly having undiagnosed AHP.

## Introduction

The growing adoption of the electronic health record (EHR) worldwide has created new opportunities for leveraging EHR data for other, so called *secondary* purposes, such as clinical and translational research, quality measurement and improvement, patient cohort identification and more [1]. One emerging use case for leveraging of EHR data is to detect undiagnosed rare diseases. Although there is no absolute definition of a rare disease, the US Rare Diseases Act of 2002 defines rare diseases as those that occur in fewer than 200,000 patients worldwide, and the National Organization for Rare Disorders (NORD,) registry lists more than 1,200 diseases. Others have noted that the true number of rare diseases is unknown, and have called for more research to define them [2].

Rare diseases can be difficult to diagnose because their infrequent occurrence may result in primary care physicians not considering them in diagnostic workups [3]. They also often have general presentations with diffuse symptoms, as well as genetic components which may require specialized testing. This lack of timely diagnosis may lead to both physical and emotional suffering as patients remain undiagnosed for prolonged periods. Additionally, a lack of accurate diagnoses increases economic burden to healthcare systems as patients continue to receive inadequate and/or inappropriate treatment. Some informatics researchers have used EHR data to detect rare diseases, such as cardiac amyloidosis [4], lipodystrophy [5], and a large collection of different diseases [6, 7].

One rare disease that may be amenable to EHR-based detection is acute hepatic porphyria (AHP). AHP is a subset of porphyria that refers to a family of rare, metabolic diseases characterized by potentially life-threatening acute attacks and, for some patients, chronic debilitating symptoms that negatively impact daily functioning and quality of life [8–12]. During attacks, patients typically present with multiple signs and symptoms due to dysfunction across the autonomic, central, and peripheral nervous systems. The prevalence of diagnosed

symptomatic AHP patients is ~1 per 100,000 [13]. Due to the nonspecific symptoms and the rare nature of the disease, AHP is often initially overlooked or misdiagnosed. A U.S. study demonstrated that diagnosis of AHP is delayed on average by up to 15 years [14].

AHP is predominantly caused by a genetic mutation leading to a partial deficiency in the activity of one of the eight enzymes responsible for heme synthesis [11]. These defects predispose patients to the accumulation of neurotoxic heme intermediates aminolevulinic acid (ALA) and porphobilinogen (PBG) when the rate limiting enzyme of the heme synthesis pathway, aminolevulinic acid synthase 1 (ALAS1), is induced [10, 14]. Gene mutations causing the disease are mostly autosomal dominant, however the disease has low penetrance (~1%) and many specific mutations have not been identified [15]. Furthermore, families carrying the gene may have few or only one affected member. Therefore, family history can be a poor diagnostic tool for this disease. The preferred diagnostic procedure for AHP is biochemical testing of random/spot urine for ALA, PBG, and porphyrin [16, 17].

Historically, treatment of AHP has predominantly focused on avoidance of attack triggers, management of pain and other chronic symptoms, and treatment of acute attacks through the use of Panhematin® (hemin for injection) [18]. Panhematin was FDA approved in 1983 for the amelioration of recurrent attacks of acute intermittent porphyria (AIP) temporally related to the menstrual cycle in susceptible women after initial carbohydrate therapy is known or suspected to be inadequate.

Recently, a new drug Givlaari® (givosiran), for subcutaneous injection has been approved by the FDA for the treatment of adults with AHP. Givosiran is a double-stranded small interfering RNA (siRNA) molecule that reduces induced levels of the protein ALAS1. A Phase 1 trial has been published [19] and a Phase 3 randomized control trial has shown this therapy to be effective in reducing the occurrence of acute attacks and impacting other manifestations of the disease [20].

The goal of this study was to apply machine learning and knowledge engineering to a large extract of EHR data to determine whether the combined approach could be effective in identifying patients not previously tested for AHP who should receive a proper diagnostic workup for AHP.

## Materials and methods

### Dataset

Oregon Health & Science University (OHSU) is the only academic medical center in Oregon and is thus a referral center for rare diseases like AHP. The OHSU Research Data Warehouse (RDW) is a research data "honest broker" service that provides EHR data to researchers, with appropriate IRB approval. The investigators have an ongoing institutional review board (IRB) approval to use an extract from the Oregon Health & Science University (OHSU) EHR research data warehouse (RDW) for a series of patient cohort identification projects. For this research, the patient cohort to identify was defined as those patients who have a documented clinical history of AHP, or a clinical history indicating that AHP diagnostic testing may be appropriate.

A large dataset of approximately 200,000 patient records was requested from the RDW, complete as of the data pull date in March 2019, including over 30 million text notes plus other document types. The data set goes back to the start of OHSU using the Epic EHR system in January, 2009. These records consist of all patients who had more than one primary care health care visit at our institution. Each patient record was represented as a collection of documents of types given in Table 1. Patient records could include zero or more documents of each type.

**Table 1. Electronic Health Record (EHR) document types used in this research.**

| EHR Document Record Type | Description of Document |
|---|---|
| Administered Medications | Medications given to patient during a hiospital stay or ambulatory encounter. |
| Current Medications | The concomittent medications a patient is taking, as documented by providers during encounters. |
| Demographics | Patient demographic information |
| Encounter Diagnosis | The diagnoses and diagnostic codes assigned to a patient ambulatory encounter. |
| Hospital Encounters | Patient-level hospital admission information including times and billing codes. |
| Lab Results | Results of ordered lab tests including order time. |
| Medications Ordered | Medications ordered by for patients by clinicians during an encounter. |
| Microbiology Results | Results of microbiology lab tests in text form. |
| Notes | All types of clinical text including progress notes and discharge summaries. |
| Problem List | The concomittent list of active medical issues for a patient, as documented by providers during encounters. |
| Procedures Ordered | Procedures ordered by clinicians for patients during an encounter. |
| Lab Result Comments | Non-numerical, text portion, if any for results of lab tests. |
| Surgeries | Description of surgeries performed on patient at hospital in both text and coded forms. |
| Vitals | Documentation of vital values such as heartrate, blood pressure, weight, and temperature. |

To insure an adequate sample size to make predictive models robust, we enriched the data set for possible AHP by adding records from an additional 5,571 patients who met one or more of the following case-insensitive criteria (see Table 2):

- Diagnosis including the wildcard search term "porph*" in the diagnosis name

- Medication including the wildcard search term "hemin*" in the medication name

- Procedure including the wildcard search term "porph*" in the procedure name

- Clinical or result note including the wildcard search term "porph*" in the note text

**Table 2. Electronic Health Record (EHR) total document and unique patients counts of porphyria codes and mentioned in text notes or label tests.** Counts shown here are out of a total of 347,709,284 individual EHR documents and 204, 413 total unique patient records.

| Code | Total Documentsts | Total Patientsents |
|---|---|---|
| ICD9 277.1 | 3879 | 308 |
| E80.0 Hereditary erythropoietic porphyria | 472 | 37 |
| E80.1 Porphyria cutanea tarda | 783 | 77 |
| E80.20 Unspecified porphyria | 2010 | 247 |
| E80.21 Acute intermittent (hepatic) porphyria | 1016 | 47 |
| E80.29 Other porphyria | 109 | 24 |
| E80.4 Gilbert syndrome | 3197 | 366 |
| E80.6 Other disorders of bilirubin metabolism | 9502 | 2308 |
| E80.7 Disorder of bilirubin metabolism, unspecified | 75 | 58 |
| Patients with porphyria mentioned in a lab test: | 359 | 175 |
| Searching field NOTE_TEXT for term porphyria: | 14353 | 3012 |

These 5,571 patient records were pulled from the RDW at the same time and in the same format as the 200K patients. There may have been some overlap between this set of patients and the 200K patients, before this data was merged into a single data set. However, all records were grouped by patient and an individual patient was only counted as a single sample in the merged data set.

To develop a gold standard for the data, a medical student (MN), overseen by clinical experts among the rest of the authors, conducted a chart review to identify patients with a confirmed diagnosis of AHP. We manually reviewed all the patients with the ICD-10-CM code E80.21 (Acute intermittent [hepatic] porphyria) in their record, looking for positive confirmation of AHP either through a lab test or a specific comment in a progress note. This process yielded 30 positive cases from the 47 coded for E80.21. As OHSU is the only academic medical center in Oregon and is thus a referral center for rare diseases like AHP, this may explain why the number of identified AHP patients in our database was higher than that which would be expected based on the global prevalence of AHP. For the remaining 17 records, we could not confirm by chart review the diagnosis of AHP. This may be due to the code being attached to the patient based on an encounter to rule out AHP, inaccurate past medical history data, or a charting error. For these 17 patients no additional information supporting the AHP diagnosis was found in the notes, clinical tests or medication records and the only evidence of AHP was an ICD-10-CM code at one place in the medical record.

The rest of the records were then assumed to be negative for AHP for the purposes of statistical analysis and machine learning. The data set consisted of the positive records plus the presumed negative records. The entire data set was used for statistical analysis and training the machine learning models, the final goal of which was to identify the presumed negative records which are actually likely to be positive.

We then deconstructed each patient record into a number of features to be used for machine learning. Structured data fields were encoded directly with the entire field content used as the feature. Free-text fields were parsed into unigrams and bigrams.

All features were labeled with their source document fields. This enabled, for example, diagnosis names in ICD-10-CM code fields in the problem list to be distinguished from the same text appearing in free text notes. Feature values were encoded as the number of occurrences in the entire record for the patient. A summary of the types and counts of documents in the data set is shown in Table 3.

## Feature selection and machine learning methods

Features to be included in the machine learning model were selected by performing univariate logistic regression analysis of the entire feature set, using the confirmed AHP patients as positive samples and the rest of the data set as negative samples.

For each document type, the 100 top features were chosen, ranked by odds ratio, having a p-value < 0.01 and occurring in at least 4 positive case patient records. This statistical criteria was used to establish which data elements had a significant relationship between the outcome variable, which was the presence, or not, of a confirmed diagnosis of AHP. Univariate analysis was performed so that individual variables could be analyzed for statistical significance and manually reviewed independently to create a smaller starting set for multivariate machine learning. Requiring that included features have at least four positive case patient records was chosen as a filter to strike a balance between only keeping the most common features, and keeping thousands of rare features requiring manual review that were unlikely be helpful in a generalized model.

**Table 3. Summary of document types and counts used in the EHR data set for this research.**

| Document Type | Patients | Encounters | Records | Median | Max |
|---|---|---|---|---|---|
| Current Medications | 187724 | N/A | 99602443 | 89 | 57406 |
| Demographics | 204413 | N/A | 204413 | 1 | 1 |
| Encounter Attributes | 204412 | 19589057 | 19589057 | 43 | 3335 |
| Encounter Diagnoses | 202843 | 10113657 | 52295188 | 69 | 27215 |
| Hospital Encounters | 145551 | 1163284 | 1163284 | 3 | 520 |
| Lab Results | 172795 | 2012185 | 58386934 | 84 | 27384 |
| Ordered Medications | 190256 | 3964120 | 15155203 | 23 | 7041 |
| Microbiology Results | 54798 | 145528 | 1988429 | 5 | 5174 |
| Notes | 204161 | 10014987 | 28938900 | 56 | 14933 |
| Problem List | 181221 | N/A | 1737749 | 6 | 204 |
| Procedures Ordered | 198833 | 5129756 | 19501225 | 31 | 35364 |
| Result Comments | 131104 | 896896 | 1542279 | 4 | 1765 |
| Surgeries | 44238 | 78403 | 83535 | 1 | 54 |
| Vitals | 199971 | 3500418 | 18268032 | 24 | 9442 |
| Administered Medications | 100565 | 349332 | 17160858 | 17 | 53178 |
| Ambulatory Encounters | 204235 | 12091755 | 12091755 | 27 | 1991 |

From these several hundred features, a manual review process was performed to ensure that none of these features were directly connected to a diagnosis of AHP, mention of AHP in the record, or treatment of AHP. This was done by inspection. This process eliminated all text features mentioning any bigram of "acute hepatic porphyria," medications such as hematin, and laboratory codes that in the OHSU system represented tests specifically for the diagnosis of porphyria.

The remaining features were then evaluated by using them in a machine learning model and scoring the model using 5 repetitions of 2-fold cross-validation. Several SVM kernel functions were tested including linear, polynomial degree 2, and the radial basis function (RBF), random forests, Adaboost, J48, and several topologies of Neural Network. Two normalization encoding methods were tried as well, binary, linear and log normalizing feature occurance counts beween 0.0 and 1.0.

After algorithm selection, a second round of feature screening was performed. Any features with non-zero algorithm weights were removed if any direct connection to AHP could be established. This was performed by close scrutiny and discussion with our clinical expert for each feature. This second pass incorporated a higher level of clinical expertise than the first pass. It was performed after filtering by machine learning weights in order to reduce the burden of manual chart review on our clinical expert.

## Machine learning for AHP prediction and evaluation methodology

A final trained model using the features selected was created by training the selected algorithm with chosen parameter settings on the entire data set. This model was then applied back to the entire data set in order to create an AHP prediction score for each patient. The classifier margin distance was taken as the prediction score. Standard SVM boundary settings were used, keeping samples scores inside the boundary region within the interval [-1, +1].

The patient prediction scores were then analyzed. To keep the manual chart review process manageable, we could not review every patient. We decided to review the top scoring 100

cases manually from each of two subsets of the general population. An alternative method would be a randomly selected subset of patients for chart review. However, because AHP is such as rare disease, the probability of finding even a single positive case with random sampling would is very small, about 0.05%.

The first reviewed subset of 100 patients were those with no mention of porphyria in their chart, no related ICD-9-CM or ICD-10-CM codes, and no porphyria specific lab test. We selected the top scoring 100 patients that met these criteria. This represents the most important target population for our project–patients with persistent symptoms that have not had AHP considered and tested to rule it in or out as a diagnosis. Manual review of these cases is intended to demonstrate the potential of our proposed approach to identify potential cases of AHP that would benefit from diagnostic testing and follow up.

The second reviewed subset of 100 patients were those <u>with</u> a mention of porphyria in the text notes in their chart, but no related ICD-9-CM or ICD-10-CM diagnosis codes, and no porphyria-specific lab test. These are patients where porphyria may have been considered by the clinician, or may have been tested at another health care facility with unavailable records, or may have been a work up in progress. Manual review of these cases was intended to discern the clinical face validity of the algorithmic predictions, that is, the high scoring patients in this group score high because the algorithm is paying attention to some of the same non-AHP-specific clinical symptoms and other variables as the clinician. While the manual review of these patients was primarily intended for gaining insight into how the algorithm was scoring patients with porphyria mentioned in the charts, based on the manual review some patients who may benefit from diagnostic testing could be found.

A clinically trained reviewer assessed the patients' records in these two non-overlapping subsets for symptom patterns consistent with acute hepatic porphyria (AHP). The reviewer was blinded to the model features. Clinical notes were searched for the 'classic triad' of AHP symptoms: abdominal pain, central nervous system abnormalities, and peripheral neuropathy [21]. In addition, any report of pain was assessed, and searches were also conducted for the highest incident AHP symptoms: abdominal pain, vomiting, constipation, muscle weakness, psychiatric symptoms, limb, head, neck, or chest pain, hypertension, tachycardia, convulsion, sensory loss, fever, respiratory paralysis, diarrhea [21]. All major comorbidities were also reviewed and documented, as well as alternative diagnoses to explain AHP symptom profiles.

The 100 patients with no mention of porphyria in their EHR record were classified into one of three categories: *AHP diagnostic testing likely indicated*, *AHP diagnostic testing possibly indicated*, and *AHP diagnostic testing unlikely indicated*. To be classified as *likely*, symptoms had to be present in all three categories of the 'classic triad', without a cause identified in the EHR, and with a substantial history of symptoms. To be classified as *possibly*, symptoms had to be present in at least one of the three categories, without a cause documented and with a substantial history. Patients were classified as *unlikely* if their symptoms could be explained by another diagnosis, or if they did not have a strong AHP symptom profile.

The 100 patients who did have a mention of porphyria in their clinical notes were classified into one of five categories of AHP status based on chart review and details in the clinical notes: *AHP already suspected*, *AHP already suspected but ruled out*, *diagnostic testing likely indicated but AHP not suspected*, *unlikely AHP*, and *AHP diagnosis mentioned in notes*. A patient was classified as *AHP already suspected* if there was any level of AHP suspicion mentioned in their clinical notes, without a formal diagnosis or lab test. *AHP already suspected but ruled out* was assigned if there was a suspicion of AHP in the note, but had been ruled out, usually by negative lab tests. These lab tests were only documented in the note, since we excluded patients from this subset who had lab tests in the laboratory data itself. *Diagnostic testing likely indicated but AHP not suspected* was assigned if there were symptoms present in at least one of the

three triad categories, without a cause, but no suspicion of AHP mentioned in the notes. For these patients the clinical notes contained the string 'porph' but presence of 'porph' in the clinical note was not related to suspicion of AHP. *Unlikely AHP* was assigned if AHP type symptoms could be explained by another diagnosis, or there was not a strong AHP symptom profile. Finally, patients were assigned to *AHP diagnosis* if there was any mention of an existing AHP diagnosis in the notes, even patient reported. The reasons for the presence of the string 'porph' in the clinical note for the second set of 100 patients was also reviewed and documented. Patient's categorized as *AHP already suspected* and *Diagnostic testing likely indicated but AHP not suspected* would benefit from AHP testing as they displayed suspicion of AHP or symptom complexes associated with AHP but have yet received a full diagnostic work-up.

This study protocol was approved by the OHSU Institutional Review Board (IRB00011159).

## Results

### Final selected features and machine learning cross-validation

Fig 1 shows a flowchart of the overall patient record filtering and manual review process. The process starts with 204,413 patient records, and using a combination of machine learning and structured data filtering described above, identifies 200 patients that were manually reviewed. 100 of those patients were identified as not having any mention of porphyria in the medical record and potentially could benefit from AHP diagnostic testing. The other 100 of those patients did have mention of porphyria in their medical record, but no diagnostic code for porphyria. These records were reviewed to determine the reason for the mention of porphyria and evaluate whether these reasons were consistent with the goal of the machine learning to

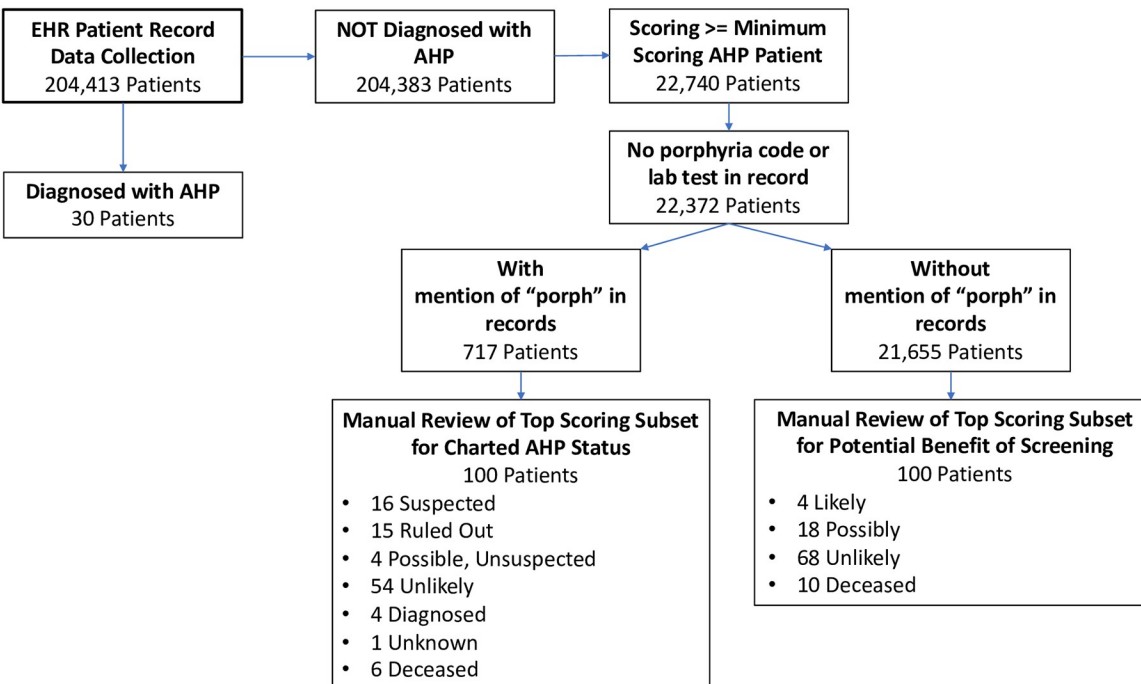

**Fig 1. Flowchart of patient data record selection.** Collection starts from full set of from full collection 204, 413 patient records and is filtered down to two sets of 100 records that were manually reviewed and characterized for 1) present indications for screening for AHP, and 2) status of AHP evaluation in the clinical notes of the record.

identify patients with symptoms and other clinical features consistent with a possible porphyria diagnosis.

Several hundred features made it through the statistical testing and occurrence frequency filter. From these several hundred features, the manual review process reduced the set to approximately 200 features. These features were then evaluated by using them in a machine learning model and scoring the model using 5 repetitions of 2-fold cross-validation. These experiments found that an SVM with the radial basis function (RBF) kernel scored best for the ranking metrics AUC and average precision. The other machine learning methods explored failed to perform as well as the RBF SVM. It was also determined that feature values were best encoded using log normalization, transforming feature occurrence counts into values between 0.0 and 1.0. Binary encoding, as well as linear normalization, failed to perform as well. We used the SVMLight implementation of the RBF kernel. Experimentation with cross-validation showed gamma = 0.04 to be optimal.

After algorithm selection and tuning, the second round of feature screening removed a few features that the SVM model assigned non-zero weights which were thought to be directly connected to the pre-established diagnosis of AHP by the clinical expert. For example, based on case series evidence, clinical hematology AHP specialists sometimes use cimetidine to treat AHP symptoms, as it is known to block a portion of the heme synthesis pathway as a side effect [22]. We found that cimetidine was a highly weighted feature in our initial models (due to its use by a specialist [TD] at OHSU based on case report data [22]) that had to be removed as it is given in response to AHP rather than being predictive. This process resulted in 141 total features being included in the final model.

The 141 features included in the final model are shown in S1 Table. Final feature set cross-validation performance on the entire training set is shown in Table 4.

## Application of machine learning to the full data set

The final machine learning model with the 141 features was trained on the entire data set, and this model was then applied back to the entire data set in order to provide a margin distance score for every patient.

The patient prediction scores were then analyzed. In particular, the range of scores obtained for the 30 confirmed positive training cases were compared to the rest of the patients in the data set. About 22,000 patients in the general population had scores that overlapped with those of the 30 positive patients. While this was only 10% of the patient records, it was more than could be manually reviewed.

We reviewed the top scoring 100 cases manually from each of two subsets of the general population. Out of the 100 patient charts we reviewed with no mention of porphyria, four were identified as likely to *AHP diagnostic testing likely indicated*, all without mention of porphyria in their medical record or documentation of a urine PBG test. The first patient was a male with six years of unexplained intermittent abdominal pain with nausea, vomiting, and

**Table 4. Cross-validation performance of the final feature set on the entire data set for ranking the 30 confirmed cases of porphyria higher than the general population.** SVM with radial basis function (RBF) kernel and gamma = 0.04.

| Metric | Score |
|---|---|
| AUC | 0.775 |
| Average Precision | 0.060 |
| Precision @ 100 | 0.031 |
| Log Loss | 0.404 |

diarrhea. His other conditions included complex regional pain syndrome, peripheral neuropathy, cardiac arrhythmias, panic attacks, and depression. The next patient was a female whose abdominal pain was described as 'a long standing symptom with extensive negative evaluation'. Also listed in her profile were neuralgias, hereditary small fiber neuropathy, movement disorder, fibromyalgia, migraines, palpitations, and somatization disorder. The third patient was a woman with multiple emergency department admissions for severe abdominal pain. She also had severe suicidality with a permanent tracheostomy due to a hanging attempt, borderline personality disorder, tachycardia, anxiety, saddle anesthesia, insomnia, and severe somatization disorder including a comment in her note advising not to admit the patient for only vague complaints. The fourth patient was a female with a history of abdominal pain comments in the notes describing that the etiology had not been identified for her complex symptomology which included headaches, abdominal pain, paresthesias and palpitations.

Overall, about a quarter of the 100 patients in the group without mention of porphyria had symptom profiles that were consistent with undiagnosed AHP and AHP diagnostic testing would either be likely or possibly indicated (Table 5). In this group there was no sign or suspicion of AHP by the clinician in the record. This is a much higher concentration of possible AHP patients than would be expected by chance based on the known prevlance of AHP.

Alternate explanations for characteristic AHP symptom profiles were diverse in the patient group without any mention of porphyria (Table 6). Cancers seen in this group included breast, uterine, pancreatic, cervical, leukemia and adrenal carcinoma. Other common comorbidities and conditions seen in this group included: fibromyalgia, irritable bowel syndrome, chronic fatigue, obesity, hypertension, obstructive sleep apnea, and chronic obstructive pulmonary disease. In contrast, alternate symptom profiles in the group with mention of porphyria in the notes were dominated by liver pathologies, mostly hepatocellular carcinoma.

Patients in the group *without* mention of porphyria in the medical record generally had much longer and more complicated histories compared to the other group, with 86 out of 100 having encounters spread over four years or longer. The patients *with* porphyria mentioned in the clinical notes tended to have shorter, and less complex histories (only 39 out of 100 had over 4 years of encounters), more focused on a single medical issue or set of symptoms, which may have been due to their being referral to our academic medical center from other health care sites.

There were small differences in age summary statistics between the two groups (Table 7), but notably more pediatric patients in the reviewed group with mention of porphyria found in

**Table 5. Assessment of the likelihood of undiagnosed acute hepatic porphyria based on clinical note symptom documentation.** Both groups of 100 reviewed patients are listed.

| | Acute Hepatic Porphyria? | # Patients |
|---|---|---|
| *No mention of porphyria group (n = 100)* | Diagnostic test is *Likely Indicated* | 4 |
| | Diagnostic test is *Possibly Indicated* | 18 |
| | Diagnostic test is *Unlikely Indicated* | 68 |
| | Deceased | 10 |
| *'Porph' in clinical notes group (n = 100)* | Suspected in chart | 16 |
| | Suspected, ruled out in chart | 15 |
| | Diagnostic test is *Possibly Indicated*, not suspected in chart | 4 |
| | Unlikely based on chart review | 54 |
| | Diagnosed, documented in chart | 4 |
| | Unknown, unable to determine | 1 |
| | Deceased | 6 |

**Table 6. Top alternative explanations for AHP symptom profiles seen in each group of patients.** Conditions seen in no more than one patient are not listed.

|  | Alternate AHP Symptom Explanation | # Patients |
|---|---|---|
| *No mention of porphyria group* | Surgery | 8 |
|  | Inflammatory Bowel Disease | 6 |
|  | Cancer | 6 |
|  | Cancer Chemotherapy | 5 |
|  | Gallbladder Pathology | 4 |
|  | Diabetes | 3 |
|  | Carnitine Palmitoyl Transferase Deficiency | 2 |
|  | Renal | 4 |
|  | Poly Cystic Ovarian Syndrome | 2 |
|  | Appendicitis | 2 |
|  | Mastocytosis | 2 |
| *'Porph' in clinical notes group* | Liver Pathology | 30 |
|  | Chemotherapy/Drug Side Effects | 3 |
|  | Mastocytosis | 2 |

clinical notes than those without (10 patients vs 1 patient). There were significantly more male patients found in this group too, compared to the group with no mention of porphyria (Table 8). Associated conditions for these 44 male patients were dominated by only a few diagnoses/symptom patterns: liver disease (N = 18), suspicion of porphyria (N = 11), or actinic keratosis (N = 3). In contrast, no single condition dominated the male disease distribution in the patient group without mention of porphyria in the notes.

About a third of patients in the group *with* mention of porphyria in the clinical notes had some level of suspicion and work-up for AHP documented. We also identified four patients in this group that we thought had possibly undiagnosed AHP, without suspicion documented in the notes. We labeled these patients as *Diagnostic testing likely indicated but AHP not suspected*. Three of these patients had 'porphyria' in their clinical note listed as a standard precaution for several different medications (hydrochloroquinone, ferrous sulfate), which they were taking. In fact, about two thirds of the patients with 'porphyria' in the clinic notes had other reasons, besides suspicion of AHP, for the presence of this word (Table 9). A large number of these patients were candidates for liver transplantation. Standard clinical documentation for evaluation for this procedure included a list of possible causes of liver failure, including

**Table 7. Age statistics in years for each of the two patient groups.**

|  | MEDIAN | MEAN | STANDARD DEVIATION | MIN | MAX |
|---|---|---|---|---|---|
| NO MENTION OF PORPHYRIA | 51 | 53 | 17.89 | 8 | 91 |
| 'PORPH' IN CLINICAL NOTES | 54 | 50 | 21.81 | 6 | 91 |

**Table 8. Sex distribution for each of the two patient groups.**

|  | MALE | FEMALE |
|---|---|---|
| NO MENTION OF PORPHYRIA | 25 | 75 |
| 'PORPH' IN CLINICAL NOTES | 44 | 56 |

**Table 9. Top reasons for the presence of the word 'porph' found in the clinical note.**

| More Common Reasons for 'Porph' in Clinical Notes | # Patients |
|---|---|
| Suspicion of Porphyria | 31 |
| Liver Transplant Documentation | 30 |
| Porphyria Mentioned in Treatment Precautions | 18 |
| Porphyria Diagnosis Mentioned in Notes | 4 |
| Porphyria Lab Tests Listed for Screening Physical | 3 |
| Family History of Porphyria | 5 |
| Misspelling | 2 |

protoporphyria. Porphyria was also mentioned as a precaution for certain medications or treatments given to some patients in this group, which included hydroxycholorquinone ferrous sulfate, therapeutic abortion, and UV light therapy for actinic keratosis.

## Discussion

This work identified four likely and 18 possible patients who had no mention of porphyria in their charts for whom AHP diagnostic testing could be indicated. In addition, four patients who had mention of porphyria in their charts not related to a diagnostic evaluation of the disease were also found likely to have AHP diagnostic testing indicated. This number of patients with indications for AHP diagnostic testing and possibly to-be confirmed diagnosis vastly exceeds that due to chance and surpassed our expectations. It will require clinical follow-up to determine whether these patients' symptoms are truly due to AHP or not, but the manual record review clearly demonstrates that our methodology has found patients for whom a spot urine porphobilinogen test is indicated.

Another benefit of identifying such patients is to inform local specialists of the presence of patients with rare diseases in which they have expertise. An institution-wide search for confirmed AHP patients through our targeted ICD-10-CM code search plus manual chart review identified 30 confirmed AHP patients. A majority of these patients were previously unknown to the porphyria specialist (TD) at OHSU. Identifying rare disease patients through large-scale data review in this manner can help connect them with the appropriate specialist to ensure optimal care.

Our results strongly suggest that leveraging of EHR data coupled with machine learning can be an effective method of identifying patients who should receive a diagnostic biochemical test to screen for AHP. Our approach was able to identify patients with compelling constellations of symptoms who had not be previously worked up for porphyria. It was also able to identify patients for whom porphyria had been considered without direct access to porphyria-related data elements such as hemin treatment, lab tests specific to AHP, or mention of AHP diagnosis in clinical notes.

This is especially interesting in the light that the overall cross-validation scores of the model on the data set using the known 30 AHP cases as the positive set and the rest of the data as negative training samples was not very high, with cross-validation yielding an average AUC = 0.775. This is somewhat of a lower performance figure then we initially expected. However, this task is very different from typical machine learning tasks due to the extremely rare nature of the positive AIP cases in both the training data as well as in the actual patient population. In most machine learning research, a data set is considered skewed or imbalanced if the number of positive cases is much less than 50%. A recent systematic review on imbalanced data classification cites articles investigating negative to positive case ratios of 100 to 1 as "highly imbalanced" (27, 28). For problems such as rare diseases, the imbalance ratio can be

nearly 10,000 to 1, as it is here. Lifting the predictive power to perhaps 22 in 100 manually reviewed cases is a potentially transformative level of performance.

The strongest positive predictors in the model included unexplained abdominal pain, pelvic and perineal pain, nausea and vomiting, and a number of pain and nausea medications. Frequent urinalysis was also a strong positive predictive feature, this is likely due to being associated with frequent ER visits and hospitalizations. The model relied on encoding the frequency of episodes, and not just binary presence of absence of symptoms. Indirectly, in the model this represented recurrent, undiagnosed problems consistent with AHP. Abdominal pain with unremarkable abdominal exam is reported as one of the most common presenting symptoms of AHP [21]. One recent cohort study also identifies nausea as significantly associated with recurrent attacks of AHP [23]. Acute pain management with opiods is also part of recent therapeutic recommendations [24].

As these methods are general, and not specific to AHP, they could be applicable to other rare disorders that have a constellation of recurrent symptoms as indicating features. There are likely ways to improve the machine learning approach, including the use of more advanced features that represent time, duration, and intervals, explicit coding of symptom separation and overlap, and more sophisticated machine learning algorithms specifically tailored to situations where the positive case is extremely rare. Investigation into machine learning algorithms for highly skewed data such as these is an active area of research [25].

## Conclusion

The combination of large data sets, machine learning techniques, and clinical knowledge engineering can be a powerful tool to identify patients with undiagnosed rare diseases. The use case of AHP presented here revealed four undiagnosed patients thought likely to have AHP, as well as 18 others who would likely benefit from testing. This level of precision in identifying potential cases of AHP from EHR data is much higher than would be expected by the prevalence of the disease.

Analyzing the EHR with advanced techniques such as demonstrated here points to the potential of the future of digital medicine on a population scale. Advanced approaches enabled by the wide deployment of the EHR can now be used to improve medicine and medical care in areas that have been underserved or inaccessible. Health care can be made more proactive, not simply in terms of common conditions and age or gender related screening, but for rarer conditions as well.

We plan to continue this work in several directions. First, it is essential for work like this to be deployed and evaluated in a clinical setting. An IRB-approved clinical validation study is being implemented. In this study, we will contact the primary care clinicians (PCP) of the patients where AHP diagnostic testing was found to be *likely* or *possibly* indicated. We will inform them that an algorithm based on EHR data has determined that their patient might have AHP and could benefit from a spot urine porphobilinogen, which is an is inexpensive, non-invasive and easy to perform diagnostic test. With the agreement of the PCP, we will then contact patients and offer them the test. Expert clinical consultation will be made available to the PCP for any questions they have. We will collect data on the interactions with the PCPs, the number of spot urine porphobilinogen tests administered, as well as the test results. In this manner, we will be able to study the clinical impact of our rare disease identification approach, beyond the retrospective, data-only study conducted and presented here.

Second, we will continue to refine our methods. Other machine learning algorithms, such as random forests and deep learning, may have advantages for AHP and other rare diseases. Other methods of encoding the EHR data that incorporate embeddings and temporal

representations, have been shown to demonstrate leading-edge results in other fields, such as computer vision, machine translation, and speech recognition, and may assist with rare diseases.

Finally, we will extend this methodology to other rare diseases that are difficult to diagnose, focusing on those for which effective treatments are becoming available. If the timeline for diagnosing rate conditions can be substantially reduced, there is great potential to impact patient health in a very significant manner.

## Supporting information

**S1 Table. Final 141 features selected for inclusion in the machine learning model to predict acute hepatic porphyria.** Features are scored by number of occurrances in an individual patient medical record, and then normalized.
(DOCX)

## Author Contributions

**Conceptualization:** Aaron M. Cohen, Steven Bedrick, John J. Ko, Jigar J. Amin, Alex J. Wei, William Hersh.

**Data curation:** Aaron M. Cohen, Steven Chamberlin, Thomas Deloughery, Michelle Nguyen, William Hersh.

**Formal analysis:** Aaron M. Cohen.

**Funding acquisition:** Aaron M. Cohen, Thomas Deloughery, Stephen Meninger, John J. Ko, Jigar J. Amin, Alex J. Wei, William Hersh.

**Investigation:** Aaron M. Cohen, Stephen Meninger.

**Methodology:** Aaron M. Cohen, Thomas Deloughery, William Hersh.

**Project administration:** Jigar J. Amin.

**Resources:** Stephen Meninger.

**Software:** Aaron M. Cohen, Steven Bedrick.

**Supervision:** Aaron M. Cohen, John J. Ko, William Hersh.

**Validation:** Aaron M. Cohen, Steven Chamberlin, Thomas Deloughery, Stephen Meninger, William Hersh.

**Visualization:** Aaron M. Cohen.

**Writing – original draft:** Aaron M. Cohen, Steven Chamberlin, William Hersh.

**Writing – review & editing:** Aaron M. Cohen, Steven Chamberlin, Steven Bedrick, Stephen Meninger, John J. Ko, Jigar J. Amin, Alex J. Wei, William Hersh.

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
