## [Decision Letter · Decision Letter 0]

5 May 2020

PONE-D-20-10338

Detecting Rare Diseases in Electronic Health Records Using Machine Learning and Knowledge Engineering: Case Study of Acute Hepatic Porphyria

PLOS ONE

Dear Dr. Cohen,

Thank you for submitting your manuscript to PLOS ONE. After careful consideration, we feel that it has merit but does not fully meet PLOS ONE’s publication criteria as it currently stands. Therefore, we invite you to submit a revised version of the manuscript that addresses the points raised during the review process.

We would appreciate receiving your revised manuscript by Jun 19 2020 11:59PM. To enhance the reproducibility of your results, we recommend that if applicable you deposit your laboratory protocols in protocols.io, where a protocol can be assigned its own identifier (DOI) such that it can be cited independently in the future. For instructions see: http://journals.plos.org/plosone/s/submission-guidelines#loc-laboratory-protocols

We look forward to receiving your revised manuscript.

Kind regards,

Sreeram V. Ramagopalan

Academic Editor

PLOS ONE

Journal Requirements:

1. Thank you for inlcuding your competing interests statement; "I have read the journal's policy and the authors of this manuscript have the following competing interests:

GIVLAARI is a product of Alnylam. GIVLAARI is a prescription medicine used to treat acute hepatic porphyria (AHP) in adults."

We note that you received funding from a commercial source:Alnylam Pharmaceuticals, Inc.

Reviewers' comments:

Reviewer's Responses to Questions

**Comments to the Author**

1. Is the manuscript technically sound, and do the data support the conclusions?

Reviewer #1: No

2. Has the statistical analysis been performed appropriately and rigorously? 

Reviewer #1: I Don't Know

3. Have the authors made all data underlying the findings in their manuscript fully available?

Reviewer #1: Yes

4. Is the manuscript presented in an intelligible fashion and written in standard English?

Reviewer #1: No

5. Review Comments to the Author

Reviewer #1: The abstract for this article is clear and concise, however the methods are unclear and difficult to follow. The methods section should provide a clear progression of logic as to how the study was conducted, ideally in linear, step-by-step fashion. Results, including from data cleaning, chart review and model parameter selection should not be included in the methods section. While the authors note that this article is a case study, the methods should nonetheless follow an a priori format. Any testing, manual review and experimenting on model structure, test statistics, etc. should be noted in the methods (ideally with a brief rationale or reference), and the results from these steps presented in the results. I believe this article can be improved, as the authors have clearly performed a considerable amount of analyses to understand prognostic features in EHR data for AHP, but without a clearly-outlined methodology I cannot evaluate the results.

6. PLOS authors have the option to publish the peer review history of their article (what does this mean?). If published, this will include your full peer review and any attached files.

Reviewer #1: No

---

## [Author Response · Author response to Decision Letter 0]

15 May 2020

Reviewer comments and our responses are given in our response letter and more conveniently formatted than are shown here.

While this is important background it is not clear if this paragraph is needed in the paper, other than noting the diagnostic/prognostics should rely on biomarker and other lab tests rather than family history. Consider removing, or condensing.

This paragraph of text is important to provide the patient disease context for our work, and provides additional clinical and genetic background to orient readers who may not have expertise about this disease, such as informaticians and machine learning researchers. The difficult diagnosis of AHP is in part due to the disease low penetrance and inconsistent appearances in families even though AHP and related diseases are mostly autosomal dominant. We therefore would like to keep the paragraph that is there now, as it really does not substantially lengthen the paper.

Recommend adding the number of patients with ICD-10 code E80.21.

This has been done.

Unique patients, or unique records/document counts? And if document counts, is this the number of unique documents with a specific code? Please clarify.

Total number of EHR records? Please clarify.

We have modified the table and caption to make these points clear.

This section is better-suited under the methods section below. Please update.

Moved as requested.

What is the start date of the data pull? How historical is the cohort?

This information has been added.

Typo? This sentence is a little confusing. Consider revising to "... adequate sample size to make predictive models robust..."

Revised as suggested.

Was this a wildcard text search? Please clarify

These are wildcard search terms, clarified in the text as requested.

You state "high likelihood" but below you note the chart review looked for a positive confirmation of AHP. It sounds like you are in fact confirming AHP through manual chart review.

This is correct. Thank you for identifying this confusion. We have revised the text to:

To develop a gold standard for the data, a medical student (MN), overseen by clinical experts among the rest of the authors, conducted a chart review to identify patients with a confirmed diagnosis of AHP.

The remaining 17 records? Please specify.

Added clarifying text:

For the remaining 17 records, we could not confirm by chart review the diagnosis of AHP. This may be due to the code being attached to the patient based on an encounter to rule out AHP, or a charting error. For these 17 patients no additional information supporting the AHP diagnosis was found in the notes, clinical tests or medication records and the only evidence of AHP was a code in the problem list or encounter diagnosis. 

Results, not methods

Results of model building, not methods.

The corresponding text has been moved to the results section, and the results section reorganized to incorporate the new text.

Model? Spelling?

Thank you for finding this error. Changed word to “algorithm”.

What is a source document? The location the field is derived in the EHR? Wouldn't that location depend on the underlying EHR structure? And why is the source document location important?

Yes, the source document is dependent upon the underlying structure of the EHR, and of our data warehouse as well. As the EHR itself is a hierarchical patient-oriented database, and our RDW is a relational database extract of that, we have no choice but to treat the records in units corresponding to the structure of the extract. These mappings between the EHR that clinicians use and the data extracts available to investigators is a common situation. The source document types correspond to units of observation common in documenting clinical care electronically. Our feature set provides both the source document and specific data field used in the model in order to provide as much information as possible to anyone trying to repeat our work and perform a similar mapping with their own EHR data. We have tried to make this more clear both in the descriptions, tables, and supplementary data.

There is no mention of constructing a training dataset in this section until the very end.

Thank you for pointing this out. We have added text to clarify how the data was used:

The rest of the records were then assumed to be negative for AHP for the purposes of statistical analysis and machine learning. The data set consisted of the positive records plus the presumed negative records. The entire data set was used for statistical analysis and training the machine learning models, the final goal of which was to identify the presumed negative records which are actually likely to be positive.

Why four patients? What was the rationale for this threshold?

Added text:

Requiring that included feature have at least four positive case patient records was chosen as a filter to strike a balance between only keeping the most common features, and keeping thousands of rare features requiring manual review that were unlikely be helpful in a generalized model.

What is the manual review process? Why not simply exclude features for EHR records that also have a corresponding AHP diagnosis, mention or treatment?

We could not exclude features as suggested since this criterion would not remove all the biased features and it may remove some associated unbiased features that could be useful.

Added: This was done by inspection using clinical domain knowledge.

How is this process different from the previous "manual review process"? Also, wouldn't the first review (if manual) have identified these same AHP-correlated features?

We needed a second pass, which included a clinical porphyria expert, to ensure that we did not miss any features that were biased by clinical pre-existing knowledge of a diagnosis of porphyria for the patient.

Added text:

This second pass incorporated a higher level of clinical expertise than the first pass. It was performed after filtering by SVM weight in order to reduce the screening load on our clinical expert.

I would expect the results section to begin with this number, highlighting the total number of patients in the entire dataset, then the final number of patients used for subsequent analyses.

Moved this text to the beginning of the results section.

General comment on all tables- please update the tables so they share the same format throughout the paper (e.g. font, font size, bold use, number formats).

We have reformatted the tables to use a consistent style.

Total number of EHR records? Please clarify.

Total number of EHR documents and patient records added to caption for Table 2.

Unique patients, or unique records/document counts? And if document counts, is this the number of unique documents with a specific code? Please clarify.

Clarified in table caption and column headings.

Please spell out the document types. The current list appears to be table names from the database itself. For example, "current_medications" should be renamed "Concomitant Medications" or "Poly-Pharmacy". "demographics" should be "Patient Demographics". I also recommend providing a brief description of these fields, as some readers may not be as familiar with traditional EHR domains. 

I recommend including standard deviation with any results presenting Mean.

Finally, be sure to format the table numbers (some rows appear to have comma delimiters, others do not).

Table 3 document type names changed to correspond with the document types in Table 1. Reformatted numbers to not use commas.

Table has been reformatted to be consistent and use full document names. Data dictionary definitions of the document types has been added to Table 1 to describe what is in these documents. Mean has been removed as table is too wide with the additions and larger font. Median and max remain and are sufficiently informative for this purpose.

Please provide either a data dictionary with descriptions for each feature, or update this table with descriptions of each feature. The current format requires the reader to assume what each feature represents based on the feature dataset name, but formal descriptions would provide more explicit clarity for the reader.

Table has been reformatted and extended to include data descriptions.

---

## [Decision Letter · Decision Letter 1]

26 May 2020

PONE-D-20-10338R1

Detecting Rare Diseases in Electronic Health Records Using Machine Learning and Knowledge Engineering: Case Study of Acute Hepatic Porphyria

PLOS ONE

Dear Dr. Cohen,

Thank you for submitting your manuscript to PLOS ONE. After careful consideration, we feel that it has merit but does not fully meet PLOS ONE’s publication criteria as it currently stands. Therefore, we invite you to submit a revised version of the manuscript that addresses the points raised during the review process.

We look forward to receiving your revised manuscript.

Kind regards,

Sreeram V. Ramagopalan

Academic Editor

PLOS ONE

Reviewers' comments:

Reviewer's Responses to Questions

**Comments to the Author**

1. If the authors have adequately addressed your comments raised in a previous round of review and you feel that this manuscript is now acceptable for publication, you may indicate that here to bypass the “Comments to the Author” section, enter your conflict of interest statement in the “Confidential to Editor” section, and submit your "Accept" recommendation.

Reviewer #1: (No Response)

2. Is the manuscript technically sound, and do the data support the conclusions?

Reviewer #1: Yes

3. Has the statistical analysis been performed appropriately and rigorously? 

Reviewer #1: Yes

4. Have the authors made all data underlying the findings in their manuscript fully available?

Reviewer #1: Yes

5. Is the manuscript presented in an intelligible fashion and written in standard English?

Reviewer #1: Yes

6. Review Comments to the Author

Reviewer #1: The revision has addressed many of the initial comments. I have provided additional comments in the attached document for your review. There are a few structural changes I'd like to recommend to strengthen the clarity of the manuscript, as well as answer questions surrounding the methods and results.

7. PLOS authors have the option to publish the peer review history of their article (what does this mean?). If published, this will include your full peer review and any attached files.

Reviewer #1: No

---

## [Author Response · Author response to Decision Letter 1]

16 Jun 2020

Here is the plain text of the response to reviewer comments as presented in the cover letter. The formatting is easier to follow in the attached cover letter.

June 16, 2020

Dear PLOS One Editors:

Please find attached a re-revised draft of our manuscript “Detecting Rare Diseases in Electronic Health Records Using Machine Learning and Knowledge Engineering: Case Study of Acute Hepatic Porphyria”. We have made additional changes based on the second round of reviewer feedback. Please thank the reviewer(s) for their helpful comments and identification of points of confusion and suggestions for clarifying the manuscript further.

We provide detailed point-by-point responses to the reviewer’s comments below.

Sincerely,

Aaron M. Cohen, MD MS

Professor

Department of Medical Informatics and Clinical Epidemiology

Oregon Health & Science University

Portland, Oregon USA 97239

 

Reviewer comments are shown in bold, our responses are given immediately afterword in plain font.

2020-05-18 18:04:31

The background information on rare diseases in general and AHP in particular is very good, however the link between rare disease identification in EHRs and the machine learning approach noted in the abstract should be strengthened. Specifically, a final paragraph highlighting the gap in current research in this area (e.g. where is current research in EHR/AHP/machine learning lacking?) , as well as the goal of this study as it related to these gaps in the research (e.g. why is this study being conducted? What gaps in the sciene will this study fill?). 

Thank you for this suggestion. We agree that the abstract does not make this connection clear. We have extended the last paragraph in the abstract to make these connections clearer. We would like to add more detail in the area but we are at the 500-word limit of the abstract.

Conclusions

The application of machine learning and knowledge engineering to EHR data may facilitate the diagnosis of rare diseases such as AHP. Further work will recommend clinical investigation to identified patients’ clinicians, evaluate more patients, assess additional feature selection and machine learning algorithms, and apply this methodology to other rare diseases. This work provides strong evidence that population-level informatics can be applied to rare diseases, greatly improving our ability to identify undiagnosed patients, and in the future improve the care of these patients and our ability study these diseases. The next step is to learn how best to apply these EHR-based machine learning approaches to benefit individual patients with a clinical study that provides diagnostic testing and clinical follow up for those identified as possibly having undiagnosed AHP.

2020-05-18 18:05:23

Consider moving IRB statement to the very end of methods section.

We have moved the IRB statement to the end of the methods section as suggested.

2020-05-18 23:36:04

I recommend making the dataset section the last section in the methods. Present the Machine learning methodology first, then feature selection, then the data.

Because the feature encoding and selection method descriptions, as well as our cohort analysis method, rely somewhat on the descriptions of the data in the dataset section, we think that it is clearer to keep the order as it is.

2020-05-18 18:43:58

This belongs at the end of the introduction, not the methods section.

Moved to the end of the Introduction as suggested.

2020-05-22 23:13:54

Where do these 5,571 patients come from? Are they a subset of the 200k from RDW, or some other EHR data source? Please specify.

They come from the same data source and were pulled from the RDW at the same time.

To clarify this we have added the text:

These 5,571 patient records were pulled from the RDW at the same time and in the same format as the 200K patients. There may have been some overlap between this set of patients and the 200K patients, before this data was merged into a single data set. However, all records were grouped by patient and an individual patient was only counted as a single sample in the merged data set.

2020-05-25 14:54:31

Why univariate? Given many of these features may have correlation, why not a multivariate model? Please provide rationale.

Added an explanatory sentence in this paragraph:

For each document type, the 100 top features were chosen, ranked by odds ratio, having a p-value < 0.01 and occurring in at least 4 positive case patient records. This statistical criteria was used to establish which data elements had a significant relationship between the outcome variable, which was the presence, or not, of a confirmed diagnosis of AHP. Univariate analysis was performed so that individual variables could be analyzed for statistical significance and manually reviewed independently to create a smaller starting set for multivariate machine learning. Requiring that included features have at least four positive case patient records was chosen as a filter to strike a balance between only keeping the most common features, and keeping thousands of rare features requiring manual review that were unlikely be helpful in a generalized model.

2020-05-22 23:18:13

Consider revising to "burden of manual chart review"

Changed as suggested.

2020-05-22 23:19:31

An alternative/complimentary method could be a randomly selected subset of patients for chart review

Thank you for the suggestion. We have added the following text as a rationale.

An alternative method would be a randomly selected subset of patients for chart review. However, because AHP is such as rare disease, the probability of finding even a single positive case with random sampling would is very small, about 0.05%.

2020-05-22 23:44:43

What was the decision boundary for the model?

We have added a sentence stating the boundary size:

Standard SVM boundary settings were used, keeping samples scores inside the boundary region within the interval [-1, +1].

2020-05-22 23:46:56

The patients were sorted by classifier margin distance and the first 100 in each group were selected?

Yes, that is correct.

2020-05-22 23:39:47

How is this model automated? It seems there is considerable manual processes involved.

We agree, ‘automated model’ is not the right term here. Changed to ‘our approach’.

2020-05-22 23:36:40

The studies cited here do not perform the same type of analysis, making comparisons to the author's AUC irrelevant. I recommend deleting, or revising with specific AUC results from studies predicting rare diseases.

Removed the comparison text and revised the paragraph as follows:

This is especially interesting in the light that the overall cross-validation scores of the model on the data set using the known 30 AHP cases as the positive set and the rest of the data as negative training samples was not very high, with cross-validation yielding an average AUC = 0.775. This is somewhat of a lower performance figure then we initially expected. However, this task is very different from typical machine learning tasks due to the extremely rare nature of the positive AIP cases in both the training data as well as in the actual patient population. In most machine learning research, a data set is considered skewed or imbalanced if the number of positive cases is much less than 50%. A recent systematic review on imbalanced data classification cites articles investigating negative to positive case ratios of 100 to 1 as “highly imbalanced” (27, 28). For problems such as rare diseases, the imbalance ratio can be nearly 10,000 to 1, as it is here. Lifting the predictive power to perhaps 22 in 100 manually reviewed cases is a potentially transformative level of performance. 

2020-05-22 23:47:48

Are there population-based studies of patients with AHP that identify these conditions/diagnoses as predictors as well? Anderson (2019) is cited above and may also be applicable here. This would strengthen the authors' results.

Added references to Anderson (2019) and other studies with some similar findings to the discussion. Thank you for the suggestion.

2020-05-22 23:33:37

Could

Thank you. Changed ‘should’ to ‘could’.

2020-05-22 23:33:04

This section reads more like a grant application. I recommend deleting, or summarizing more broadly as areas to continue the research.

We have made a small addition to the text to clarify our intent, but we do not think substantial changes to these three paragraphs are necessary. We believe it is important for us to convey these specific future directions for this work. They provide an important perspective on where research like this should head after the initial phase described in this paper. The text also highlights the importance of clinical evaluation and follow up of informatics and machine learning research to produce improvement in patient care.

2020-05-22 23:31:03

Could tables 5-9 be combined into one table?

We think it would be very dense and confusing to combine all these tables into one table. The individual captions help to explain the tables and what they present in a focused manner. Also the font size would have to be shrunk and that would make readability difficult.

Also, tables 5-6 present patient subgroups on the horizontal rows, while tables 7-8 present the patient subgroups in the vertical columns. I reccomend picking one or the other and applying to all tables.

We have changed tables 7 and 8 to present patient subgroups in horizontal rows.

2020-05-22 23:28:29

Each patient group (e.g. clinical notes and no mention of porphyria)?

Changed to ‘each’ as suggested.

2020-05-22 23:26:35

Each patient group (e.g. clinical notes and no mention of porphyria)?

Changed to ‘each’ as suggested.

2020-05-22 23:29:10

Generally, standard deviation is presented with mean

We have included a column for standard deviation.

---

## [Editor Report · Decision Letter 2]

18 Jun 2020

Detecting Rare Diseases in Electronic Health Records Using Machine Learning and Knowledge Engineering: Case Study of Acute Hepatic Porphyria

PONE-D-20-10338R2

Dear Dr. Cohen,

We’re pleased to inform you that your manuscript has been judged scientifically suitable for publication and will be formally accepted for publication once it meets all outstanding technical requirements.

Kind regards,

Sreeram V. Ramagopalan

Academic Editor

PLOS ONE
---

## [Editor Report · Acceptance letter]

23 Jun 2020

PONE-D-20-10338R2 

Detecting Rare Diseases in Electronic Health Records Using Machine Learning and Knowledge Engineering: Case Study of Acute Hepatic Porphyria 

Dear Dr. Cohen:

I'm pleased to inform you that your manuscript has been deemed suitable for publication in PLOS ONE. Congratulations! Your manuscript is now with our production department. 

Kind regards, 

on behalf of

Dr. Sreeram V. Ramagopalan 

Academic Editor

PLOS ONE